# Potential G-quadruplexes and i-Motifs in the SARS-CoV-2

**Efres Belmonte-Reche**[1]*, **Israel Serrano-Chacón**[2], **Carlos Gonzalez**[2], **Juan Gallo**[1], **Manuel Bañobre-López**[1]

**1** Advanced (Magnetic) Theranostic Nanostructures Lab, INL-International Iberian Nanotechnology Laboratory, Braga, Portugal, **2** Instituto de Química Física'Rocasolano', CSIC, Madrid, Spain

* efres.belmonte@inl.int

**Data Availability Statement:** All results and the R package are available from the URL https://github.com/EfresBR/G4iMGrinder.

**Funding:** M.B-L & J.G: a. Grants: 1. NORTE-01-0145-FEDER-000019, 2. NORTE-01-0145-FEDER-

## Abstract

Quadruplex structures have been identified in a plethora of organisms where they play important functions in the regulation of molecular processes, and hence have been proposed as therapeutic targets for many diseases. In this paper we report the extensive bioinformatic analysis of the SARS-CoV-2 genome and related viruses using an upgraded version of the open-source algorithm G4-iM Grinder. This version improves the functionality of the software, including an easy way to determine the potential biological features affected by the candidates found. The quadruplex definitions of the algorithm were optimized for SARS-CoV-2. Using a lax quadruplex definition ruleset, which accepts amongst other parameters two residue G- and C-tracks, 512 potential quadruplex candidates were discovered. These sequences were evaluated by their *in vitro* formation probability, their position in the viral RNA, their uniqueness and their conservation rates (calculated in over seventeen thousand different COVID-19 clinical cases and sequenced at different times and locations during the ongoing pandemic). These results were then compared subsequently to other *Coronaviridae* members, other Group IV (+)ssRNA viruses and the entire viral realm. Sequences found in common with other viral species were further analyzed and characterized. Sequences with high scores unique to the SARS-CoV-2 were studied to investigate the variations amongst similar species. Quadruplex formation of the best candidates were then confirmed experimentally. Using NMR and CD spectroscopy, we found several highly stable RNA quadruplexes that may be suitable therapeutic targets for the SARS-CoV-2.

## Introduction

The severe acute respiratory syndrome coronavirus 2 (SARS-CoV-2) is a positive-sense single-stranded RNA virus from the *Betacoronavirus* genus, within the *Coronaviridae* family of the *Nidovirales* order. Although it is believed to have originated from a bat-borne coronavirus [1–5], the SARS-CoV-2 can spread between humans with no need of other vectors or reservoirs for its transmission. The virus is responsible for the ongoing COVID-19 pandemic that has caused hundreds of thousands of deaths, millions of infected and a disastrous strain on the economy of most countries and citizens worldwide.

The origin of the virus has been traced back to the Chinese city of Wuhan, where the first cases of infected individuals were reported amongst the workers of the Huanan Seafood

031142 and 3. 0624_2IQBIONEURO_6_E. b.
Funders: 1. 2014-2020 North Portugal Regional
Operational Program (NORTE 2020) and the
European Regional Development Fund (ERDF), 2.
the Fundação para a Ciência a a Tecnoloxía (FCT),
ERDF and NORTE 2020, 3. 2014-2020 INTERREG
Cooperation Programme Spain–Portugal
(POCTEP). c. URLS: 1. https://norte2020.pt,
https://ec.europa.eu/regional_policy/en/funding/
erdf/, 2. https://www.fct.pt, https://norte2020.pt,
https://ec.europa.eu/regional_policy/en/funding/
erdf/, 3. https://interreg.eu/programme/interreg-
spain-portugal-poctep/ The funders had no role in
study design, data collection and analysis, decision
to publish, or preparation of the manuscript. C.G: a.
Grants: BFU2017-89707-P b. Funders: Spanish
Ministry of Science, Innovation and Universities
(MCIU) c. URL: https://www.ciencia.gob.es d. The
funders had no role in study design, data collection
and analysis, decision to publish, or preparation of
the manuscript.

**Competing interests:** The authors have declared
that no competing interests exist.

Market [6, 7]. This wet exotic animal market, where wild animals including bats and pangolins are sold and prepared for consumption, offers ample opportunities for pathogenic bacteria and viruses to adapt and thrive [8, 9]. Such circumstances led Cheng and colleagues to predict the current pandemic back in 2007 [10]. In their own words: "*the presence of a large reservoir of SARS-CoV-like viruses in horseshoe bats, together with the culture of eating exotic mammals in southern China, is a time bomb. The possibility of the re-emergence of SARS and other novel viruses from animals or laboratories and therefore the need for preparedness should not be ignored*".

SARS-CoV-2 has now become a global problem. In this current scenario, the scientific community is playing a fundamental role in minimizing the number of victims. Their work includes, to name a few, the development of fast and reliable detection methods, the identification of therapeutic targets within the virus, and the development of active drugs and vaccines to cure and to prevent infections, respectively.

G-Quadruplexes (G4s) and i-Motifs (iMs) have been proposed as therapeutic targets in many disease aetiologies. G4s are Guanine (G) rich DNA or RNA nucleic acid sequences where successive Gs stack in a planar fashion via Hoogsteen bonds to form four-stranded structures, stabilized by monovalent cations [11]. iMs on the contrary, are Cytosine (C)-rich regions that fold into tetrameric structures of stranded duplexes [12–14]. These are sustained by hydrogen bonds between the intercalated nucleotide base pairs $C \cdot C^+$ when under acidic physiological conditions.

The importance of these genomic secondary structures has been abundantly studied during the last years [15–20]. They have been found to be regulatory elements in the human genome implicated in key functions such as telomere maintenance and genome transcription regulation, replication and repair [21]. G4 structures have also been identified in fungi [22–25], bacteria [26–30] and parasites [31–36]. Their occurrence are known in many viruses that infect humans as well. These include the HIV-1 [37–39], Epstein-Barr [40, 41], human and manatee papilloma [42, 43], herpes simplex 1 [44, 45], Hepatitis B [46], Ebola [47] and Zika [48] viruses. Here they can regulate the viral replication, recombination and virulence [32, 49, 50].

iMs have been less studied in general, especially outside of the human context. With regards to viruses, Ruggiero *et al.* recently published the formation of an iM in HIV-1 [51], whilst we reported the presence of the known cMyb.S [52] iM within the Epstein-Barr virus [53]. Despite the lack off reports, iMs are interesting potential therapeutic targets for viruses. For example, the *in silico* analysis of the rubella virus revealed an extremely dense genome of potential iMs (density as counts per genomic length) that surpassed its human counterpart by over an order of magnitude [53]. In the same study, other viruses such as the measles and hepacivirus C presented potential iMs densities similar to the human genome.

In this work, we wished to contribute to the ongoing research efforts related to the COVID-19 pandemic by investigating SARS-CoV-2 for the presence of quadruplex structures. With this aim, we analysed the prevalence, distribution and relationships of Potential G4 Sequences (PQS) and Potential iM Sequences (PiMS) in its genome. These PQS and PiMS have been evaluated according to their potential to form quadruplex structures *in vitro* and localization within the genome. The presence of confirmed quadruplex-forming sequences and the candidate's frequency, uniqueness and conservation rates between 17312 different SARS-CoV-2 clinical cases were also analyzed. The study of the SARS-CoV-2 and its quadruplex results were expanded to integrate the *Coronaviridae* family, Group IV of the Baltimore classification and the entire virus realm, as to allow a wider range of interpretation. With all this information at hand, our final objective was to identify biologically important PQS and PiMS candidates in the virus. To substantiate our bioinformatic analysis, we analysed experimentally some of these sequences by CD and NMR spectroscopies. Our *in vitro* results confirmed the formation

of stable quadruplexes that can form in the viral genome, suggesting that they may be suitable targets for new therapeutic or diagnostic agents [50, 54]. Hence, our analysis of the SARS-CoV-2, and by extension of the entire virus realm, may provide useful insights into using quadruplex structures as targets in future anti-viral treatments.

## Materials and methods

### G4-iM Grinder and G4-iM Grinder' parameter configuration

In this work, we have used an upgraded version of the G4-iM Grinder package (GiG, https://github.com/EfresBR/G4iMGrinder) for the analysis of all viruses (S1 File, section 1). GiG is an R-based algorithm that locates, quantifies and qualifies PQS, PiMS and their potential higher-order versions in RNA and DNA genomes [53]. We retrieved the SARS-CoV-2's reference sequence (GCF_009858895.2) from the NCBI database [55]. We also downloaded those of 18 other viruses which can cause mortal illness in humans, including six other pathogenic Coronavirus, as comparison (S1 File, section 2).

As a workflow, we applied the functions *GiG.Seq.Analysis* (to study their G- and C-run characteristics), *G4iMGrinder* (to locate quadruplex candidates) and *G4.ListAnalysis* (to compare quadruplex results between genomes) from the GiG package to all the viruses. The 'size-restricted overlapping search and frequency count' method (Method 2, M2A and M2B) was used to locate all the candidates. Then, these PQS and PiMS were evaluated by the presence within of *in vitro* confirmed G4 or iM sequences, their frequency of appearance in the corresponding genome, and their probability of quadruplex-formation score (as the mean of G4Hunter [56] and the adaptation of the PQSfinder algorithm [57]). To compare between virus species, we calculated the density of potential quadruplex sequences per 100000 nucleotides (Density $= 100000 \times \frac{Number\ of\ candidates}{Genome\ Length}$).

We previously saw that viruses have a wider-range of PQS and PiMS densities than that of the human, fungi, bacteria and parasite genomes [53]. Some were totally void whilst others were very rich in candidates. So, we explored different quadruplex definitions to determine the most useful configurations for the analysis of the viruses at hand. These different definitions control the characteristics of what the algorithm considers a quadruplex. They include the acceptable size of G- or C-repetitions to be considered a run, the acceptable amount of bulges within these runs, the acceptable loop sizes between runs, the acceptable number of runs to constitute a PQS or PiMS, and the total acceptable length of the sequence (Fig 1A). A flexible configuration of quadruplex definitions will detect larger amounts of candidates at the expense of requiring more computing power and accepting sequences that are more ambiguous in forming quadruplex structures *in vitro* (as determined by their score; with longer loops, smaller runs, more bulges and more complementary G/C %, **Fig 1B**). More constrained definitions result in the opposite. Hence, for the analysis, we chose three different configurations: a *Lax* configuration (which accepts run bulges and longer ranges of runs, loops and total sizes), the *Predefined* configuration of the package (which restricts sizes but still accepts run bulges), and the original *Folding Rule* [58, 59] (which restricts length and does not accept run bulges) (**Fig 1C** Left).

Then, we calculated the PQS and PiMS densities of each virus to allow a direct size-independent comparison between them all (**Fig 1D**), and filtered the results by their *in vitro* probability of formation score. The |score| filters were set to 20 and 40 to allow us the study of both the medium (PQS score $\geq$ 20; PiMS score $\leq$ -20) and the high probability candidates (PQS score $\geq$ 40; PiMS score $\leq$ -40; **Fig 1B**) within the results. These score filters are important because they qualify the sequences and grant specificity to the results of GiG's extremely

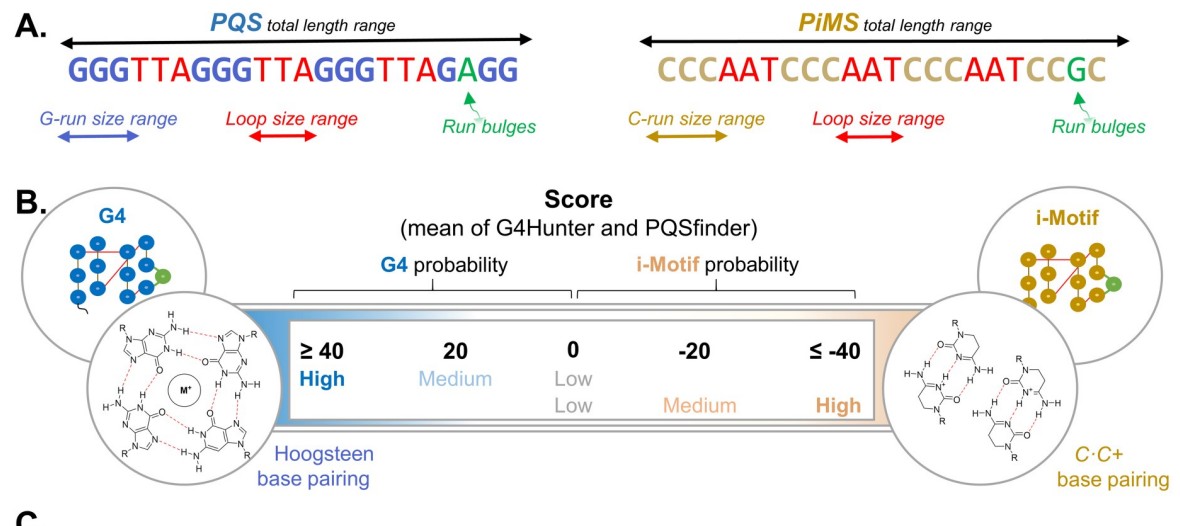

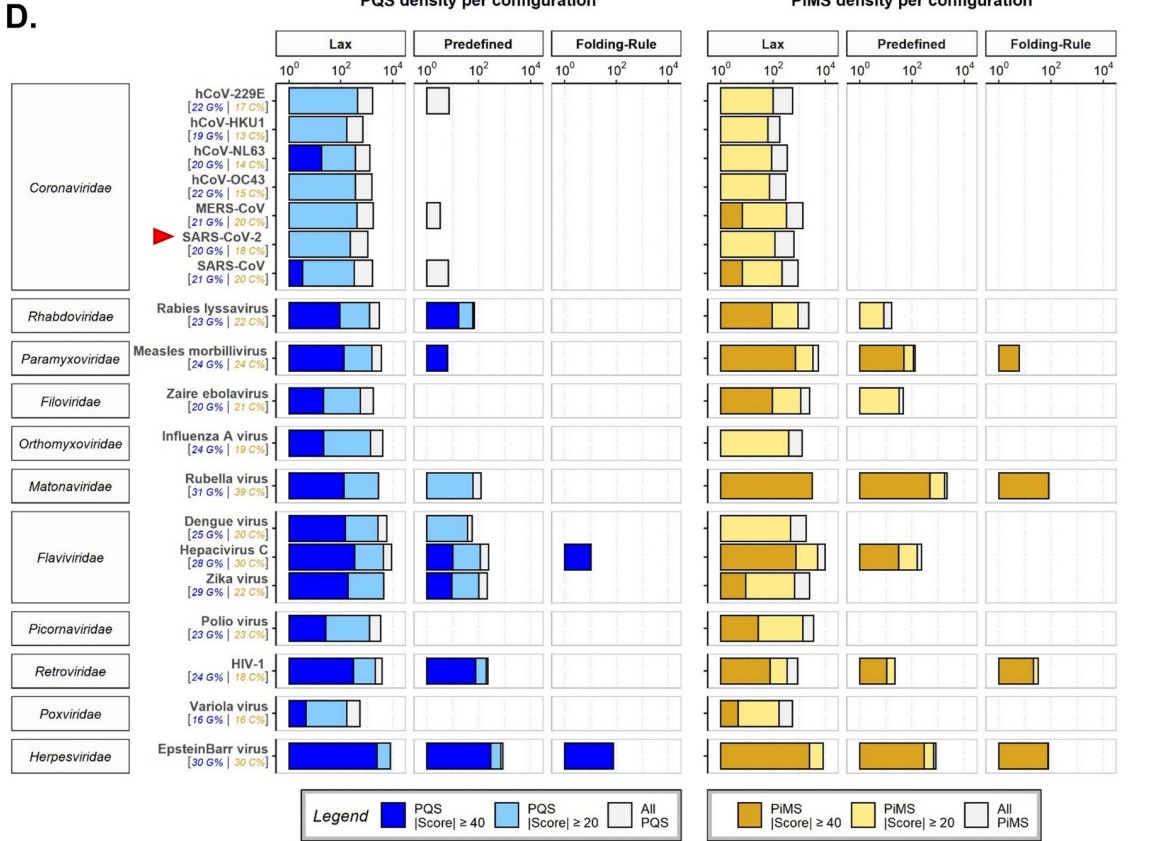

**Fig 1.** **A**. Results with G4-iM Grinder depend on the quadruplex definitions introduced to the algorithm. Sizes of G- or C-runs, loops and the entire sequence, together with an acceptable number of bulges within the runs are part of the definitions. **B**. The structures found with GiG under the definitions proposed by the user can be evaluated for their *in vitro* probability of formation. More positive scores mean that the sequence is more capable of forming G4s, whilst more negative values mean that it is more capable of forming iMs. **C**. Left, Quadruplex definitions used by GiG's search engine in this work. **C**. Right, Total results found within the SARS-CoV-2 by configuration and score criteria. **D**. PQS and PiMS densities (per 100000 nucleotides) found per different configuration and score criteria for 19 viruses. The G and C content (as a percentage) is shown under each virus. X scale is in logarithmic scale (base 10). Results are categorized by their |score|: intense colours (blue for PQS, yellow for PiMS) are the most probable to form *in vitro* (|score| $\geq$ 40), lighter bars are the density of structures with at least a |score| $\geq$ 20 and grey bars are the densities without the score filter.

flexible search engine (which was designed solely for sensitivity), as highlighted by the results of a recent review [60].

For the viruses analysed, the best configuration to obtain significant number of candidates was the *Lax* set-up. This was also relevant for the reference genome of the SARS-CoV-2 (**Fig 1C** Right). Given the small size of the viral genomes, the increase in computational power was deemed acceptable and hence, we established this *Lax* configuration as the default configuration for all posterior searches with GiG. Although some authors have reported the unfeasibility of forming iMs with tracks of only two C [19], such statement has been rebutted later [61], allowing the use of this configuration also for potential iMs.

The search was then expanded to 17312 different SARS-CoV-2 genomes sequenced during the pandemic (from December-2019 to January-2021, by different laboratories worldwide and downloaded from the GISAID database [62]), other *Coronaviridae* family members and the entire virus realm (6678 other viruses) using the methodologies described previously and in the S1 File, section 1. To validate the *in silico* findings, the most interesting candidates were selected and confirmed by NMR and CD spectroscopy.

## *in silico* methodology

To analyse these genomes, we employed the workflow described in the G4-iM Grinder' parameter configuration section of the manuscript using the *Lax* parameter configuration. We investigated the biological features potentially affected by candidates using the function *GiG.df.GenomicFeatures* of the GiG package. The conservation of each PQS and PiMS found in the reference genome was calculated as {Conservation (%) = $100 \times \sum Ng^+/\sum Ng$} where $Ng$ is the number of genomes, and $Ng^+$ is the number of genomes with the PQS or PiMS candidate. The genomic pairwise alignments, used to study the similarity between viruses and detect PQS and PiMS variations between species, were done using the *pairwiseAlignment* function (global alignment type) from the *Biostrings* package in the Bioconductor repository. We calculated the divergence from the reference genome per clade (or lineage) as, {$Divergence = (|\overline{N^{\circ}_{Clade/Lineage}}|S.| \geq 20 - N^{\circ}_{ref}|S.| \geq 20|) + \overline{N^{\circ}_{Lineage\ Variants}}|S.| \geq 20$} where $\overline{N^{\circ}_{Clade/Lineage}}|S.| \geq 20$ is the clade/lineage's mean number of PQS or PiMS that |score| at least 20, $N^{\circ}_{ref}|S.| \geq 20$ is the number of PQS or PiMS that |score| at least 20 in the reference genome, and $\overline{N^{\circ}_{Lineage\ Variants}}|S.| \geq 20$ is the mean number variants of PQS or PiMS that |score| at least 20 per Lineage. To compare potential quadruplex presence and prevalence between genomic groupings (species, families, groups and the entire virus realm), we calculated also the genomic density of several arguments. These were calculated using the *GiGList.Analysis* function of the GiG package (density per 100000 nucleotides). The arguments were the density of results (PQS and PiMS), density of results with |score| filters (with at least 20 or 40), density of already confirmed sequences that form G4 or iM within, and uniqueness (as {Uniqueness (%) = $100 \times \sum Ns^{f=1}/\sum Ns$} where $Ns$ is the number of sequences, and $Ns^{f=1}$ is the number of sequences with a frequency of appearance of 1 in its respective genome). For the G- and C-runs density analysis of the viruses, we used the function *GiG.Seq.Analysis* from the GiG package. The arguments here were: densities of runs with different

sizes (two or three to five long G- or C-runs) and with different bulges per run (zero and/or one). All of these results can be found in the S1 File, section 5.

## Candidate selection

PQS and PiMS candidates were selected according to their potential to form quadruplex structures *in vitro*, uniqueness, frequency of appearance, conservation between 17312 different SARS-CoV-2 clinical case genomes, confirmed quadruplex presence and localization within the genome.

## NMR experiments

Oligonucleotides (0.3 mM) for NMR experiments were purchased from IDT, and suspended in 200 μl of $H_2O/D_2O$ 9:1 in 25 mM $KH_2PO_4$ and 25 mM KCl buffer, pH 7. Samples at acidic pH were prepared by adding aliquots of concentrated HCl. Spectra were acquired on Bruker Avance spectrometers operating at 600 MHz, and processed with Topspin software. Experiments were carried out at temperatures ranging from 5.1 to 45˚C and pH from 5 to 7. NOESY spectra in $H_2O$ were acquired with a 150 ms mixing time. Water suppression was achieved by including a WATERGATE module in the pulse sequence prior to acquisition.

## Circular Dichroism (CD)

Circular dichroism (CD) studies were performed on a JASCO J-810 spectropolarimeter using a 1 mm path length cuvette. Spectra were recorded in a 320–220 nm range at a scan rate of 100 nm min$^{-1}$ and a response time of 4.0 s with four acquisitions recorded for each spectrum. Data were smoothed using the means-movement function within the JASCO graphing software. Melting transitions were recorded by the monitoring the decrease of the CD signal at 264 nm. Heating rates were 30˚C/h. Transitions were evaluated using a nonlinear least squares fit assuming a two-state model with sloping pre- and post-transitional baselines. Oligonucleotide solutions for CD measurements were prepared at the same buffer conditions as the NMR experiments. Oligonucleotide concentration was of 50 μM.

## Results and discussion

A detailed analysis of the results of SARS-CoV-2, *Coronaviridae* family and the entire virus realm with G4-iM Grinder can be found in the S1 File, Section 3.

## G4-iMGrinder and settings

The genome of the SARS-CoV-2, and that of many other viruses, were analysed with G4-iM Grinder in search off potential quadruplex (both G4 and iM) therapeutic targets. To do so, we first expanded G4-iM Grinder's quadruplex identification and characterization repertoire with two new functions, *GiG.Seq.Analysis* and *GiG.df.GenomicFeatures*. Other functions such as *G4iMGrinder* and *GiGList.Analysis* were upgraded to better analyse and summarise the quadruplex results obtained. Furthermore, over 2800 quadruplex-related sequences were searched for in the literature and included in G4-iM Grinder's database to rapidly identify confirmed G4s and iMs within all results.

An initial study of the SARS-CoV-2 genome and 18 other pathogenic viruses revealed the special characteristics that need to be considered for quadruplex-related examinations in these organisms. For most, the original folding rule (which accepts no bulges within the runs and is very constrained in its quadruplex definitions) and the predefined parameters of G4-iM Grinder (which allows more liberty by accepting bulges and longer loops) are too strict to find

associated runs that can give rise to quadruplexes. Although other organisms such as *Plasmodium falciparum* or *Entamoeba histolytica* may be poorer in G and C content [53], the size of these genomes enables finding rich G- or C- tracks that can ultimately form potential quadruplexes. In most viruses, however, this does not take place because of the small size of the genomes (in the range of tens to hundreds thousand nucleotides versus the tens of millions for the parasites mentioned, and thousands of millions for humans). Furthermore, most of the G4s found in viruses are complex sequences, with short runs and bulges (for example, HIV-1 [37, 39] and Ebola [47]), which elude detection when following traditional quadruplex definitions. To overcome these problems, we took advantage of the great adaptability of G4-iM Grinder, and developed, tested and successfully employed a *lax* quadruplex definition configuration for the analysis. With these settings, the number of candidates found increased greatly and included the complex sequences expected in viruses, at the expense of needing more computational power.

## SARS-CoV-2

With all these updates and configurations at hand, we focused on the reference SARS-CoV-2 and located 323 PQS and 189 PiMS unique (only occurring once in the genome) sequences dispersed unevenly in its genome (**Fig 2**). 20% of these candidates had at least a medium probability of formation (|score| ≥ 20), and 7 PQS and 10 PiMS had a |score| ≥ 30 (**Fig 2D**). Candidates with at least a medium probability of formation concentrate in the N, S and especially in the orf1ab gene (in the nsp 1 and 3 regions for PQSs and in the nsp 3, 4 and 12 regions for PiMS). The orf3a, orf8 and UTR regions also presented these candidates. Other genes, such as orb7a and b, and orf10 were found totally void of them.

We calculated the SARS-CoV-2 candidate's quadruplex conservation rates and quadruplex-related region variability under three different scopes.

First, attention was focused exclusively on the virus in an intra-species analysis comprising 17312 genomes of the SARS-CoV-2 sequenced at different places and times of the pandemic. Here, we found that the least conserved candidates were located in the 5'UTR, orf1ab and N regions with conservation as low as 9.8%. On the other hand, most of the sequences analysed that |scored| ≥ 20 presented conservation rates of over 99% (46/71 PQS and 21/35). Of these, only 18 PQSs and 7 PiMSs rates surpassed that of the mean sequence identity percentage between the 17312 SARS-CoV-2 and the reference genome (99.83%). To further investigate these differences, we first identified the 5429 new PQSs and 3298 new PiMS variants that | scored| ≥ 20 amongst all the SARS-CoV-2 genomes and then associated them with the versions found in the reference genome. In this manner, we identified for one of the highest-scoring PQSs found in the N-gene (entry 7, **Fig 2D** and entry 1, **Fig 3A**) a variant with the same probability of formation (entry 3, **Fig 3A**), which is exclusive to the lineages within B.1.1/clade GR and B.1.160/clade G. These have a substitution of a C for a U in the first loop, and together with several other less frequent variations with similar modifications in the loops, partially explain its 99.08% conservation rate. Furthermore, a nearby four-membered G-run may influence this PQS, to the point of potentially being a fifth domain [63, 64] or forming an alternative G4 (entry 2, **Fig 3A**). This extra G-run is separated from the PQS by a 19-nucleotide long loop that has a conservation rate of only 35%. The most frequent variants found for this poorly conserved area were also the substitution of a C for a U, as seen before (entry 4, **Fig 3A**). Variants of lineage A/Clade S displayed a different substitution, where a C mutates to a G and becomes an additional G-run, which can further influence the PQS (entry 5, **Fig 3A**). How this affects the known activity of the PQS and the N gene is yet to be determined [65]. Variants of specific lineages with heightened quadruplex formation probability were also detected for

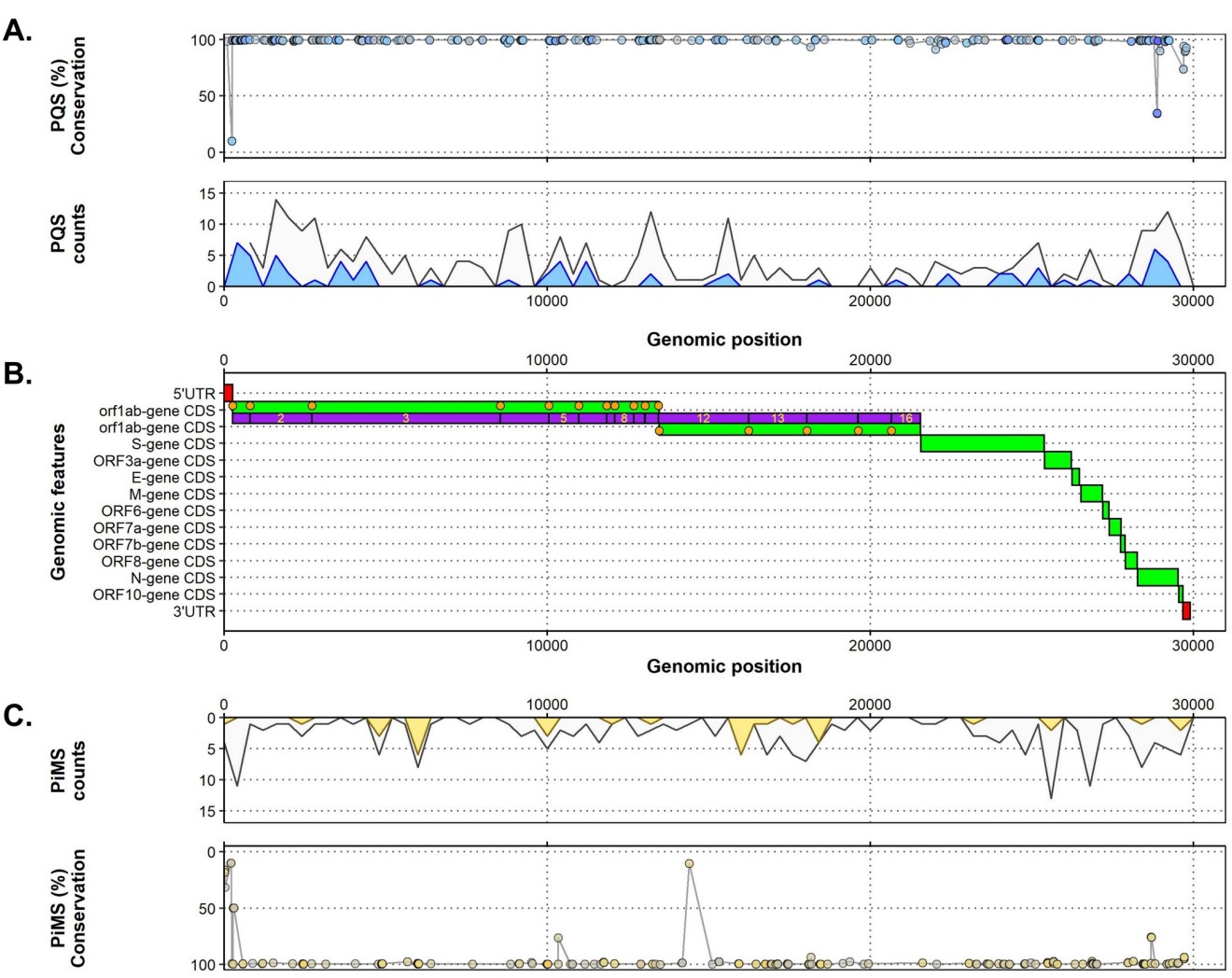

| Entry | Start | Sequence | Score | Biological feature | SARS-CoV-2 conservation (%) |
|---|---|---|---|---|---|
| 1 | 359 | GGAGACUCCGUGGAGGAGG | 30 | orf1ab - nsp1 | 99.44 |
| 2 | 370 | GGAGGAGGUCUUAUCAGAGG | 30 | orf1ab - nsp1 | 99.46 |
| 3 | 644 | GGUAAUAAAGGAGCUGGUGG | 30 | orf1ab - nsp1 | 99.93 |
| 4 | 3467 | GGAGGAGGUGUUGCAGG | 34 | orf1ab - nsp3 | 99.86 |
| 5 | 4255 | GGGUCAGGGUUUAAAUGGUUACACUGUAGAGGAGG | 31 | orf1ab - nsp3 | 99.37 |
| 6 | 13385 | GGUAUGUGGAAAGGUUAUGG | 31 | orf1ab - nsp10 | 99.92 |
| 7 | 28903 | GGCUGGCAAUGGCGG | 34 | N | 99.08 |
| 8 | 15 | CCUUCCCAGGUAACAAACCAACC | -31 | 5' UTR | 16.02 |
| 9 | 4886 | CCUACCACAUUCCACC | -36 | orf1ab - nsp3 | 99.57 |
| 10 | 6011 | CCAAACCAACCAUAUCC | -34 | orf1ab - nsp3 | 99.62 |
| 11 | 10015 | CCAACCACCACAAACC | -36 | orf1ab - nsp4 | 99.77 |
| 12 | 10015 | CCAACCACCACAAACCUCUAUCACC | -32 | orf1ab - nsp4 | 99.71 |
| 13 | 10019 | CCACCACAAACCUCUAUCACC | -32 | orf1ab - nsp4 | 99.73 |
| 14 | 15924 | CCUUCCUUACCCAGAUCC | -38 | orf1ab - nsp12 | 99.54 |
| 15 | 15924 | CCUUCCUUACCCAGAUCCAUCAAGAAUCC | -32 | orf1ab - nsp12 | 99.47 |
| 16 | 15928 | CCUUACCCAGAUCCAUCAAGAAUCC | -30 | orf1ab - nsp12 | 99.54 |
| 17 | 16750 | CCUAGACCACCACUUAACC | -32 | orf1ab - nsp13 | 99.89 |

**Fig 2. A.** Top. Percentage of conservation of each PQS found along the genome of the SARS-CoV-2. Each point represents one PQS. The PQS score is given by the fill colour of the points, where lower |scores| are greyer, and bluer points have higher |scores|. Bottom. PQS count density plot related to the genome position (counts per 200 nucleotides). Grey coloured density plots are all the results found, whilst blue density plots are the results found with at least a |score| ≥ 20. **B**. Distribution of the biological features of the SARS-CoV-2 by its genomic position. UTR regions are in red, CDS and genes region are in green, and nps of the orf1ab gene are in purple. Orange dots are mature protein regions of the CDS. **C**, Top. PiMS count density plot related to the genome position (counts per 200 nucleotides). Grey coloured density plots are all the results found, whilst yellow density plots are the results found with at least a |score| ≥ 20. Bottom. Percentage of conservation of each PiMS found along the genome of the SARS-CoV-2. Each point represents one PiMS. The PiMS score is given by the fill colour of the points, where lower |scores| are greyer, and higher |scores| are more yellow. **D**. Top scoring PQS (Score ≥ 30, entry 1 to 7) and PiMS (Score ≤ -30, entry 8 to 17) found in the SARS-CoV-2 ordered by their localization in the genome. G-runs are in blue, C-runs are in yellow, loops are in red and bulges within the runs are in green. For each entry, the biological feature column lists the genomic landmark that hosts the potential quadruplex. The percentage of conservation is also given.

several other high scoring candidates, including a PQS found in the 5'UTR area (entry 1, **Fig 2**A and entry 6, **Fig 3**A) and a PiMS in the orf1ab gene (entry 14, **Fig 2**A and entry 8, **Fig 3**A), both of which are the only results found in SARS-CoV-2 with high a probability of forming quadruplex (|score| ≥ 40).

We observed significant differences between the SARS-CoV-2 lineages and clades when considering the overall PQS differences. On the one hand, the GR clade displayed a reduced number of PQSs, PQSs that |scored| at least 20 and the least number of variants per genome analysed (**Fig 3**C). On the other hand, The S clade presented, on average, additional PQSs in their genome and a higher number of variants per genome analysed. In either case, both clades differed significantly from the reference genome, as well as amongst themselves. The rest of the clades presented fewer differences although some specific lineage aggregations (B.1.1/Clade O and B.1.1/Clade G) also displayed a lower number of PQSs overall. For PiMS, the differences between clades were smaller and more homogeneous (S1 File, section 3, **Fig 2**C).

The search was then expanded to the rest of the *Coronaviridae* family. 53 SARS-CoV-2 PQS and PiMS candidates were found in common with the SARS-CoV and/or Bat coronavirus BM48-31/BGR/2008 (Bat-CoV-BM), all of which are suspect of having bats as hosts during their evolution (S1 File, section 3, **Fig 3**). These common sequences were located in the 3'UTR, N and E genes of the SARS-CoV-2, although most were positioned in the orf1ab gene, and especially in the 5'UTR region. Paradoxically, the candidates found in the 5'UTR site (which regulates the translation of the RNA transcript) include the least conserved group of candidates of the inter-species analysis (with conservation rates as low as 9%), while also hosting a very conserved family-wise group of candidates. On the one hand, high conservation in candidates (maintained through natural selection) may be an important factor for the survival of the virus. This importance may transcend beyond the SARS-CoV-2 and into other familiar species were PQS and PiMS were found in common. On the other hand, variability in the region may also play a vital role in the ability of the virus to adapt to new hosts, situations and environments.

The highest |scoring| candidates found in SARS-CoV-2 were however not common to any other *Coronaviridae* member species. So, we investigated the differences between them through genome alignments and found that most of the sequence versions amongst species (6 out of 8) were still able to form potential quadruplex structures even with modifications. Therefore, these PQS and PiMS, although different from those in the SARS-CoV-2, maintain their potential biological role and importance.

Expanding the search for common candidates to the entire virus realm, we matched one PQS and PiMS from the SARS-CoV-2 with the potential quadruplexes found in four viruses from Group I belonging to the *Herpesviridae*, *Podoviridae* and *Siphoviridae* families (all dsDNA) which cannot be explained by the number of sequences analysed.

**A.**

| Entry | Start ref. | Sequence | Score | Biological feature | SARS-CoV-2 conservation (%) | |
|---|---|---|---|---|---|---|
| 1 | 28903 | GGCUGGCAAUGGCGG | 34 | N | 99.08 | |
| 2 | 28881 | GGGGAACUUCUCCUGCUAGAAUGGCUGGCAAUGGCGG | 28 | N | 34.81 | |
| 3 | | GGUUGGCAAUGGCGG | 34 | N | 0.25 | Clade GR; Lineage B.1.1.x & Clade G; Lineage B.1.160 |
| 4 | | GGGGAAUUUCUCCUGCUAGAAUGGCUGGCAAUGGCGG | 28 | N | 0.83 | |
| 5 | | GGGGAACUUCUCCUGGUAGAAUGGCUGGCAAUGGCGG | 32 | N | 0.03 | Clade S; Lineage A.3 |
| 6 | 359 | GGAGACUCCGUGGAGGAGG | 30 | 5'UTR | 99.44 | |
| 7 | | GGGGACUCCGUGGAGGAGG | 42 | 5'UTR | 0.05 | Clade G; B.1 Lineage & Clade S; Lineage A |
| 8 | 15924 | CCUUCCUUACCCAGAUCC | -38 | orf1ab - nsp12 | 99.54 | |
| 9 | | CCUUCCUUACCCAGACCC | -44 | orf1ab - nsp12 | 0.05 | Clade GR; Lineage B.1.1.x |

**B.**

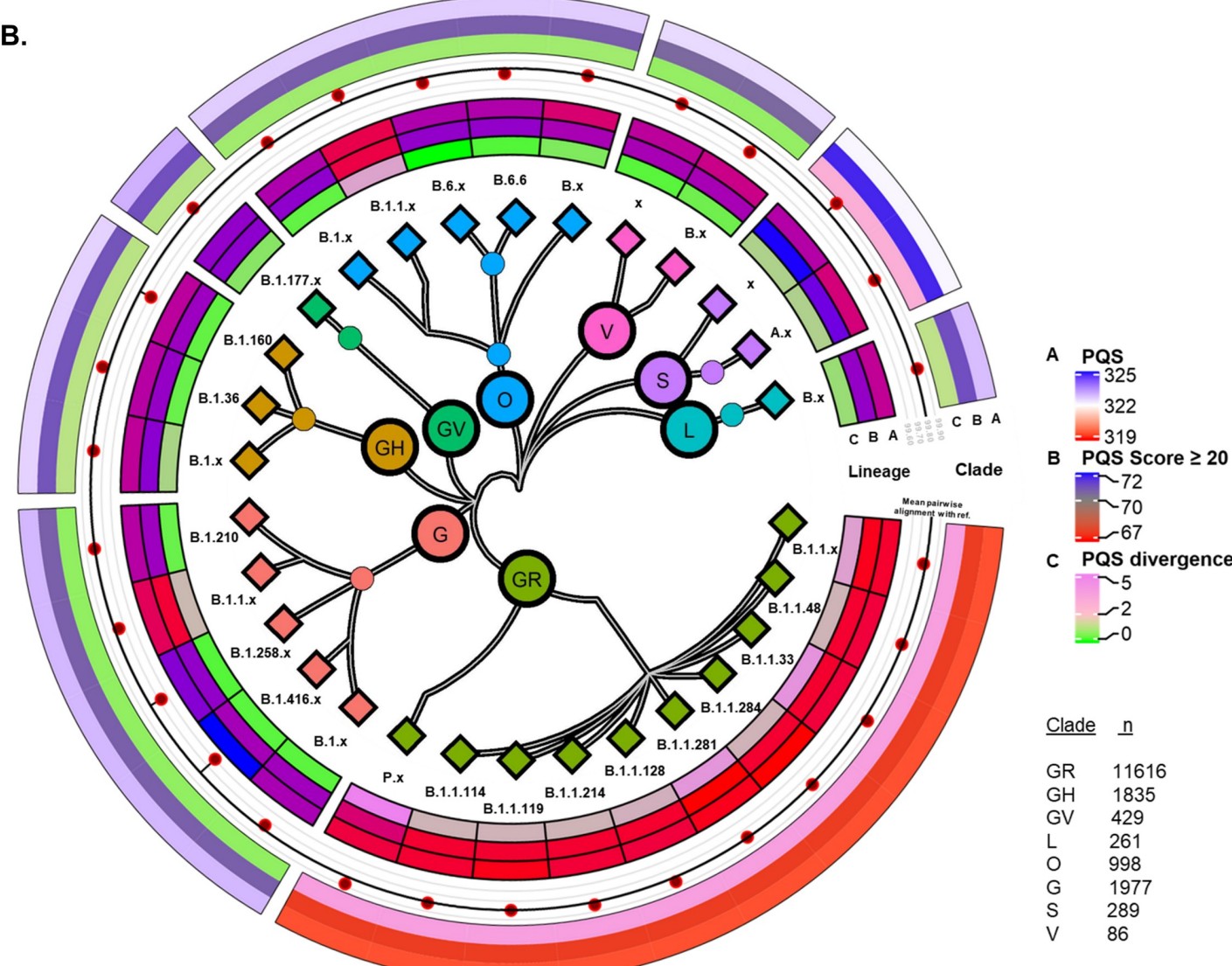

**Fig 3.** **A**. Sequences found in the SARS-CoV-2 reference genome (those with a starting position) and some of the variants identified in specific lineages for four high scoring candidates. Mutations are underlined. **B**. Centre, SARS-CoV-2 phylogenetic tree by clade and lineage of the sequences analysed. Lineages with less than 100

genomes were grouped (suffix x). Inner segment, Lineage: Mean PQS count (A), Mean PQS *count* with |score| ≥ 20 (B) and PQS divergence from the reference genome (C). Centre segment, Mean lineage percentage sequence identity with the reference genome (dots) compared to the overall mean found for the 17312 sequences analysed (black line). Outer segment, Clade: Mean PQS count (A), Mean PQS *count* with |score| ≥ 20 (B) and PQS divergence from the reference genome (C). *R-packages used*: *ggtree* [66] *and circlize* [67].

### SARS-CoV-2 and the virus realm

We analysed the entire virus realm in a similar fashion to other studies in the literature [68, 69]. However, we employed the *lax* definition of quadruplexes to detect G- and C- structures and searched for verified G4 and iM sequences already described in literature. These results were then matched and compared to SARS-CoV-2.

Whilst the SARS-CoV-2 did not present any of the published quadruplex sequences listed in the *GiG.DB* (as of V2.5.0) within its genome, other viruses including a wigeon-afflicting Coronavirus did. In the entire virus realm, 1725 viruses presented at least one confirmed G4 sequence in their genome, while 195 at least one confirmed iM sequence (the dimensional discrepancies between both results may partially be due to the difference in the number of G4 and iM entries in the database; 2568 and 283 respectively). The sheer volume of species with confirmed quadruplex structures in all groups of viruses suggests that quadruplexes may be common and necessary genomic regulatory elements for viruses, as seen in other organisms such as humans. However, the prevalence is not homogeneous and varies broadly at the group level although not that much at the family level. For example, some families like Group I's *Herpesviridae* and *Sphaerolipoviridae*, Groups IV's *Matonavirirdae* and *Flaviviridae* and Groups II's *Spiroviridae* presented the highest PQS densities; whilst Groups V's *Aspiriviridae* and *Fimoviridae*, Groups IV *Mononiviridae* and *Mesoniviridae* and Group's I *Mimiviridae* displayed the lowest. PiMS showed a similar tendency with Group I (*Sphaerolipoviridae* and *Herpesviridae*) and especially IV (*Tymoviridae*, *Matonaviridae* and *Gammaflexiviridae*) families being the densest in candidates; whilst Groups IV (*Monoviridae* and *Yueviridae*), Groups V (*Fimoviridae* and *Phasmavirirdae*) and Groups I families (*Mimiviridae*) displayed the lowest. These results indicate that viruses/families (and particularly single-stranded ones) are probably more oriented to a kind of quadruplex structure in a group/genome-type independent manner, whilst being contingent upon cation concentration and pH of the environment for formation.

Altogether, the SARS-CoV-2 genome displayed a quadruplex candidate scarcity when compared in a macroscopic perspective to the virus realm. Its PQS and PiMS densities were in the lower end of results from the *Coronaviridae* family, which itself was in the lower end of the (+) ssRNA Group IV (in an approximate ratio of 1:2:4 for PQS and 1:2:8 for PiMS). When put into the entire virus realm context, the SARS-CoV-2 PQS density was lower than 5813 other viruses analysed (out of 6680), whilst PiMS density was lower than 6125. Furthermore, when we compared the SARS-CoV-2 reference genome results with the results of five hundred randomly shuffled genomic sequences of size and composition equal to that of SARS-CoV-2, the number of candidates found in the SARS-CoV-2 was significantly lower than the mean expected number of candidates for the genome's size and composition. Whilst 362 ± 42 PQSs were expected, only 323 were found in the SARS-CoV-2. Similarly, 97 ± 22 PQSs that score over 20 and 3.0 ± 2.6 candidates that score over 40 were expected with this genomic size and composition, whilst 71 and 0 were found in the virus, respectively. For PiMS the total number of expected candidates for the SARS-CoV-2 size and genome composition was of 250 ± 33, whilst candidates that score -20 or less and -40 or less was of 60 ± 16 and 1.5 ± 1.6, respectively. However, SARS-CoV-2 presented only 189, 32 and 0 PiMSs for each of these respective groups. Although the SARS-CoV-2 genomic organization limits the number of potential

quadruplex structures almost to its minimum, other viruses with similar low quadruplex densities were identified here to possess confirmed G4 and iM sequences within, supporting the potential these structures have for targeting the SARS-CoV-2.

## Candidate confirmation *in vitro*

We, therefore, selected the best candidates to evaluate *in vitro*. NMR spectra of CoVID-RNA. G4-1 and CoVID-RNA.G4-2 exhibited imino signals in the 10.5–12.0 region, characteristic of guanine imino protons involved in G-tetrads (**Fig 4**A and 4B). In both cases, CD spectra also showed the characteristic positive band of parallel G-quadruplexes, which together with the NMR results confirmed the formation of very stable structures. The highly conserved CoVID-RNA.G4-1 located in the N-gene can possibly interact with the viral RNA packaging, transcription and replication functions [70]. In fact, it has been shown in a recent study that a known G4-ligand can interact with this sequence and reduce the expression of the N protein [65]. Although CoVID-RNA.G4-2 also formed a stable parallel quadruplex, the signals in the NMR spectra were broader than for CoVID-RNA.G4-1. This might be due to the formation of higher order structures through self-association between G-quadruplex units. CoVID-RNA. G4-2 is located in the nsp3 region of orf1ab very near its SUD domain. This area has been associated with the increased pathogenicity of the virus compared to other *Coronaviridae* that do not present it [71]. Additionally, it has been suggested that the SUD domain interacts with G-quadruplexes of the host. These results, however, open the possibility of an intrinsic gene modulation that may be linked with an increased virulence. Such a hypothesis can be extended to the SARS-CoV, as another stable PQS candidate was found in its genome in the same location (S1 File, Section 3, Fig 3B1).

For PiMS, we used NMR to confirm that the DNA version of a candidate located in the orf1ab gene of the SARS-CoV-2 and with a 99.54% conservation rate formed an iM at almost neutral pH (**Fig 4**C and S1 File, Section 3, Fig 5). However, the SARS-CoV version of the iM (which differs by one nucleotide in the first loop, from TT to TG) was unable to form even at pH 5.1. As TT base pairs are common capping positions, the substitution of the T might prevent the folding in SARS-CoV. Additionally, the presence of C in G4s lowers overall stability of the quadruplex as C can base pair with G and ultimately hinder G-quartet formation [72]. Similarly, the pairing of C with G may also impede the formation of the C-based structures. When we analysed the RNA version of the SARS-CoV-2 iM, it did not form an iM. Despite the fact that the sequences found in SARS-CoV-2 have an intermediate probability of formation, RNA iMs are known to be less stable than their DNA-versions [73]. Still, G4-iM Grinder methodology identified several more candidates with the potential to form iMs in the virus.

## PQS result comparison

The results of G4-iM Grinder were compared to other recent reports of quadruplex-related analysis in the single strand of SARS-CoV-2. QGRS mapper [74] was the main tool for the search because of its browser-based interface, its predefined capability to detect two-sized G-runs and its design that returns all the PQSs found independently of their score [65, 75–78]. Other search engines such as G4Hunter and PQSfinder automatically filter their results by their score threshold, which makes criterion optimization fundamental to successfully execute the analysis. For example, one PQSs was found with a threshold of $\geq 1.2$ and none with higher thresholds when using G4Hunter in the virus (in its scale of -4 to 4) [75]. On the contrary, 25 candidates have been reported using QGRS mapper with very small scores (mean QGRS Score of $12 \pm 5$ in QGRS mapper's scale of $\approx 0$ to 100; mean G4Hunter score of $0.6 \pm 0.2$ in G4Hunter scale). G4catchall [79], PQSfinder and QGRS mapper methodologies were also

**A.**

| | Type | Sequence | *in vitro* | Biological feature | SARS-CoV-2 conservation (%) | TAG |
|---|---|---|---|---|---|---|
| *Fig. 1.A, 7* | RNA | UGGCUGGCAAUGGCGGU | Y | N | 99.08 | CoVID-RNA.G4.1 |
| | DNA | TGGCTGGCAATGGCGGT | Y | | | CoVID-DNA.G4.1 |
| *Fig. 1.A, 4* | RNA | UGGAGGAGGUGUUGCAGGA | Y | orf1ab - nsp3 | 99.86 | CoVID-RNA.G4.2 |
| *Fig. 1.A, 14* | RNA | AUCCUUCCUUACCCAGAUCCUA | N | orf1ab - nsp12 | 99.54 | CoVID-RNA.iM.1 |
| | DNA | ATCCTTCCTTACCCAGATCCTA | Y | | | CoVID-DNA.iM.1 |
| | DNA | ATCCTGCCTTACCCAGATCCTA | N | | | SARS-DNA.iM.1 |

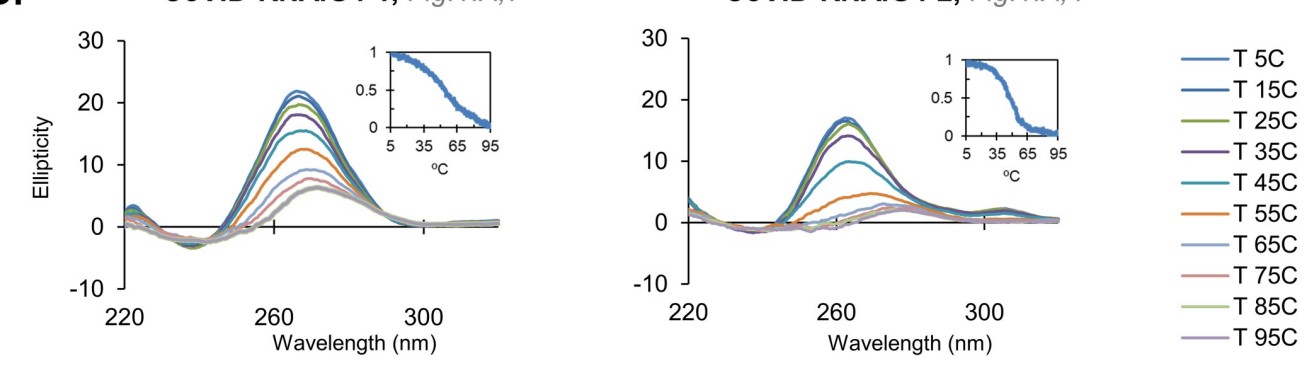

**Fig 4.  A**. The candidates examined *in vitro* through biophysical assays. The *in vitro* column states if the sequence forms a quadruplex (Y for Yes, N for No). **B**, NMR spectra of the two RNA-G4s analyzed at different temperatures (pH 7.0). **C**, NMR spectra of the DNA-iM analyzed at different temperatures (pH 5.3). **D**, CD analysis of the two RNA-G4 analyzed.

combined to select 15 PQSs, 13 of which were part of the original QGRS mapper results [80]. Except one, all of these sequences reported to date in SARS-CoV-2 have been found with G4-iM Grinder and are part of the analysis made here. These are (mainly) part of the 71 sequences with a medium probability of forming G4 (scored between 20 and 40 in G4-iM Grinder's scale). G4-iM Grinder however, found 47 extra PQSs that have not been previously reported for the SARS-CoV-2 with the same probability of forming G4s. Additionally, over 5000 different variants of these PQSs were also identified with the same probability, in the analysis of the 17312 different SARS-CoV-2 genomes.

Overall, these results complement the current knowledge we have regarding quadruplexes and the SARS-CoV-2. They also broaden the way for targeting viruses in general, and the SARS-CoV-2 in particular, through the use of these nucleic sequences as therapeutic targets in future anti-viral treatments. G4-ligands based on small molecules that can stabilize G4s have recently been proposed to be viable antivirus strategies for viruses such as Ebola, HIV and HCV (reviewed in [50]). For the SARS-CoV-2, G4-ligands have already been reported to significantly reduce protein translation levels *in vivo* and *in vitro* [65]. Another report highlighted the existing evidence indicating that helicase inhibitors may also exert antiviral activity as another therapeutic approach for SARS-CoV-2 [78].

## Supporting information

**S1 File.**
(PDF)

## Acknowledgments

The authors thank Dr. Matilde Arévalo, Rafael Ferreira and Sarah Heselden for their help regarding this topic.

## Author Contributions

**Conceptualization:** Efres Belmonte-Reche.

**Formal analysis:** Efres Belmonte-Reche, Israel Serrano-Chacón, Carlos Gonzalez.

**Funding acquisition:** Carlos Gonzalez, Juan Gallo, Manuel Bañobre-López.

**Investigation:** Efres Belmonte-Reche, Israel Serrano-Chacón, Carlos Gonzalez.

**Methodology:** Efres Belmonte-Reche.

**Software:** Efres Belmonte-Reche.

**Validation:** Efres Belmonte-Reche.

**Visualization:** Efres Belmonte-Reche.

**Writing – original draft:** Efres Belmonte-Reche.

**Writing – review & editing:** Efres Belmonte-Reche, Carlos Gonzalez, Juan Gallo, Manuel Bañobre-López.

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
