## [Decision Letter · Decision Letter 0]

14 Jan 2021

PONE-D-20-34438

Exploring G- and C-quadruplex structures as potential targets for the severe acute respiratory syndrome coronavirus 2

PLOS ONE

Dear Dr. Belmonte Reche,

Thank you for submitting your manuscript to PLOS ONE. After careful consideration, we feel that it has merit but does not fully meet PLOS ONE’s publication criteria as it currently stands. Therefore, we invite you to submit a revised version of the manuscript that addresses the points raised during the review process.

After careful review of the comments, it appears that the reviewers have a several issues with the manuscript. One of the main concerns was the use of the 'lax' setting for the G4-iM Grinder and the resulting predicition of G-quadreplexes based on this setting, as well as comparision with alternative G-quadraplex tools such as e.g. cGcC or G4Hunter.

Please address these concerns, along with the other comments made by the reviewers, in your re-submission.

We look forward to receiving your revised manuscript.

Kind regards,

Eric Charles Dykeman, Ph.D.

Academic Editor

PLOS ONE

Journal Requirements:

Reviewers' comments:

Reviewer's Responses to Questions

**Comments to the Author**

1. Is the manuscript technically sound, and do the data support the conclusions?

Reviewer #1: Partly

Reviewer #2: Partly

Reviewer #3: Partly

2. Has the statistical analysis been performed appropriately and rigorously? 

Reviewer #1: Yes

Reviewer #2: Yes

Reviewer #3: No

3. Have the authors made all data underlying the findings in their manuscript fully available?

Reviewer #1: Yes

Reviewer #2: Yes

Reviewer #3: No

4. Is the manuscript presented in an intelligible fashion and written in standard English?

Reviewer #1: No

Reviewer #2: Yes

Reviewer #3: Yes

5. Review Comments to the Author

Reviewer #1: Exploring G- and C-quadruplex structures as potential targets for the severe acute respiratory syndrome coronavirus 2

By Efres Belmonte-Reche* et al (*Corresponding author)

Submitted to PLoS One (Editorial No. PONE-D-20-34438)

General Comments

G- and C-quadruplex secondary structures (G4s and iMs, respectively) are found in the cellular genomes of many animal species and have complex roles in the regulation of metabolic pathways, including the biochemistry of telomeres and of oncogenic promoters (genome stability). Bacteria and many viruses have recently been explored for the presence of such structures in their genomes. Here, the genomes of SARS-CoV-2 isolates (of the reference strain, GCF_009858895.2, and of >3000 subsequent isolates), of other coronaviruses (CoVs) and of other (+)ssRNA viruses were investigated for the presence of such structures using the G4-iM Grinder (GiG) algorithm with some modifications. The quadruplex sequence structures identified were evaluated for the probability of occurring in infected cells, their position in the genome, and the degree of conservation in closely related viruses of the Coronaviridae family. The ‘best’ candidates for potential quadruplex formation were explored for stability by biophysical techniques (NMR, CD spectroscopy). Several relatively stable quadruplexes were identified which may be considered as possible targets for the development of novel antivirals.

The data observed are interesting. However, their presentation is rather confusing:

- The Materials and Methods section frequently refers to the Suppl. Mat. component of the manuscript for information which should be in the main text;

- The modifications of the GiG method used are not properly explained;

- The ‘detailed analysis of the Results’ is moved into Suppl. Mat. which is rather confusing and also has led to some duplications in panels of figures in the main text and in Suppl. Mat. ;

- The frequency of quadruplex structure candidates is very high at the lax quadruplex definition, but the relevance of such data in biological terms is unclear;

- The significance of differences in conservation of potential quadruplex structures is not clear;

- Text and Legend of Fig. 2 differ in the nomenclature of structures assessed;

- The findings of this manuscript are claimed to ‘greatly expand the current knowledge regarding quadruplexes and the SARS-CoV-2 in particular [refs 66-68]’ (lines 320/321), but no details are given;

- No particular information is given on compounds which could potentially react with (and disturb) quadruplex structures, using them as therapeutic targets.

Numerous clarifications are requested, and the presentation of this potentially very important information should be improved.

Specific Comments

Line

1 Reconsider title, e.g., ‘Potential G- and C-quadruplex structures of SARS-CoV-2 as potential targets for the development of antivirals’, or similar. [Since no antiviral candidates are even mentioned, consider: ‘G- and C-quadruplex structures of SARS-CoV-2 and their potential biological relevance’]

18 A short paragraph should introduce the topic.

28 … the entire realm… Please clarify. …. in common with other species… Please specify.

32 … may be suitable targets for… Please clarify (see comment above).

38 Consider citation of: Lau SKP, et al. Possible Bat Origin of Severe Acute Respiratory Syndrome Coronavirus 2. Emerg Infect Dis. 2020 Jul;26(7):1542-1547.

Andersen KG, et al. The proximal origin of SARS-CoV-2. Nat Med. 2020 Apr;26(4):450-452.

45 … pathogenic bacteria and viruses…

47 Consider citation of:

Menachery VD, et al. A SARS-like cluster of circulating bat coronaviruses shows potential for human emergence. Nat Med. 2015 Dec;21(12):1508-13.

Peck KM, et al. Coronavirus Host Range Expansion and Middle East Respiratory Syndrome Coronavirus Emergence: Biochemical Mechanisms and Evolutionary Perspectives. Annu Rev Virol. 2015 Nov;2(1):95-117.

57 … DNA or RNA [read RNA throughout the ms; ARN is the French abbreviation.]

74 Consider reading: … great potential as targets for virus inhibition…

77 Clarify the description of data from ref. [49]. Correct the site of publication to: NAR Genomics and Bioinformatics, Volume 2, Issue 1, March 2020, lqz005, https://doi.org/10.1093/nargab/lqz005

78 Support the statement by refs.

79 Rephrase sentence: … to the ongoing research efforts related to the COVID-19 pandemic by investigating SARS-CoV-2 for the presence of quadruplex structures…

84 Rephrase sentence.

111 … presence of known-to-form quadruplex structures… Please clarify.

133 and 158f. Please clarify what ‘positive’ and ‘negative’ mean in this context. Are these designations just used for presentational purposes?

154 Fig. 1A. Consider showing a quadruplex structure.

172f This summarizing referral of the reader to Suppl. Mat. is confusing. In addition, in detail there are partial duplications in figures of Suppl. Mat. and the main text. It should be considered to transfer essential components of Suppl. Mat. into the main text (including relevant figures). See comments below.

208 to 213. The meaning of this text is not clear. The concluding sentence ‘Here they may play their biological role if formed’ is speculative and should be omitted as long as no hard data are available.

220 This interpretation of Fig. 2A does not describe what is shown. Please adjust accordingly.

224 and 230. The citation of components of Suppl. Mat. is out of order and confusing.

247 Fig. 2, panel C. Explain how COVID DNA.iM-1 was obtained and what this means in context.

260f Clarify sentence.

267 … quadruplexes may be common and… for viruses to ‘live’, thrive and adapt… This statement is highly speculative and should be considered for omission.

293 and 296. Clarify: … CoVID-RNA G4-1… CoVID-RNA G4-2… The interpretation of the latter structure is highly speculative and should be omitted.

310 … the opposite but with the same effect might also be happening… Please spell out more clearly what you want to say.

318f This paragraph should contain details of how the data presented here augment those of refs. [60, 61, 63, 66-68]. Furthermore, the potential of developing novel antivirals interfering with quadruplex structure formation, should be assessed in more detail.

476 Correct ref. , see comment above.

520 Qu X is last, not first author.

Suppl. Mat.

Page

3 The new functions developed and their exact website locations should be mentioned under Methods in the main text.

4 The subheadings of Suppl. Fig. 1 could be incorporated into Fig. 1, making Suppl. Fig. 1 redundant.

5 Essential information of ‘in silico methodology’ should be in the Methods section of the main text.

6 The procedure of NMR and circular dichroism experiments should be in the Methods section of the main text.

7 A condensed version of this text should be transferred to the Methods section of the main Text.

Paragraph 2, line 3. Omit ref [2].

8f Suppl. Fig. 2 panel A duplicates Fig. 2 panel A. Suppl Fig. 2 panels B-D could remain in Suppl Mat.

11f Abbreviated versions of these data should be transferred to the main text.

14f and Suppl. Fig. 4 could remain in Suppl. Mat. in a condensed form.

18f Biophysical experiments. The Suppl. Fig. 5 duplicates data of Fig. 2. The section should be transferred to the main text and abandoned in Suppl. Mat.

21f ‘Other bioinformatics figures’. Suppl. Figures 6-11 and 13-16 are not considered to be essential. Suppl. Figs. 12 and 17 could be considered for the main text.

33f ‘Other biophysical figures’. Suppl. Figs. 18 and 19 could be considered for Suppl. Mat., preferentially Suppl. Fig. 19.

35 to 37. Condensed components of this information (including relevant additional refs.) should be transferred to Methods of the main text.

Reviewer #2: In their manuscript the authors present a genome wide screen for putative

G- and C Quadruplexes sites (PQS and PiMS, respectively) in SARS-CoV-2 and other

viral genomes. For that purpose, the authors updated the pre-existing

R-based scanning tool G4-iM Grinder.

The genome wide screen in the SARS-CoV-2 reference genome resulted in about

300 PQS and about 200 PiMS. However, when the resulting scores were filtered

by a previously suggested value of |score| > 40, none of the predicted sites

remained. Therefore, the authors concentrated on those results with |score| > 20

which consisted of 71 PQS and 32 PiMS. The authors then compared these

sites to other viral genomes within the Coronaviridae, the entire group IV

and a selection of all virus genomes to find sites conserved within different

virus species. Furthermore, the authors conducted NMR and CD experiments on

a handful of selected high-scoring PWS and PiMS found in SARS-CoV-2 reference

genome. Here they found that both of the PQS tested do form a G-quadruplex.

The single PiMS, however, did not show formation of a C-quadruplex. However,

for the latter the authors also tested a DNA variant instead and found, that

in DNA this sequence indeed adopts a C-quadruplex conformation. Interestingly,

a single point mutation in the first loop region this DNA variant disrupts

the ability to form the C-quadruplex.

Overall, the article is very well written, and the analysis performed by the

authors is comprehensive and sound. Nevertheless, there are some minor points

that require more attention or need to be described in more detail. Furthermore,

in some instances the authors draw conclusions that I can't follow or agree

with.

In particular, I have the following remarks and questions:

1. In the methods section, the authors state that in their analysis they

chose 3 different configurations of G4-iM Grinder, the lax, the default,

and the 'original folding rule'. However, all Results are only for the

lax configuration. If I understand correctly, this was the only configuration

that yielded any results for SARS-CoV-2. If none of the results presented

in this study (and the corresponding supplement), please correct the Methods

section accordingly.

Along with that, I wonder how the authors justify their threshold of |score|>20.

From the original G4-iM Grinder publication I took that |score|>40 is suggested

for good prediction performance. Here, I am missing some mroe discussion

about this lower threshold, especially in combination with the lax configuration,

as I assume that this might greatly increase the number of false positive predictions!

2. The authors should consider comparing their result of 73 PQS to those found

in another previous study by Ji et al. 2020 "Discovery of G-quadruplex-forming

sequences in SARS-CoV-2", Briefings in Bioinformatics (https://doi.org/10.1093/bib/bbaa114)

3. 3rd paragraph of the Materials and Methods section, line 125ff. The authors

state that a more flexible configuration requires 'more computing power'. Maybe,

I oversaw this but I wonder why this is the case and how the computation time

scales with the choice of the different 'flexibility' parameters. Does the

analysis scale linearly, quadratically, or even exponential in the number of

candidate sites? One can only speculate at this point. Especially, since the

underlying scoring schemes (cGcC, G4Hunter, etc) seem to be independent

on the flexibility of the sequence constraint.

4. Results and discussion SARS-CoV-2, 5th paragraph, lines 242-245, as well

as Supplement Section 3, 'SARS-CoV-2, the Virus Realm and quadruplexes'.

The authors matched PQS and PiMS found in SARS-CoV-2 in other viruses and

claim that these findings 'cannot be explained by chance'. I argue against that,

since there is no obvious relation between dsDNA viruses and the ss+virus

SARS-CoV-2. Given the probability in the order of 1e-13 as derived in the

supplement and the vast size of the entirety of viral genomes, I would expect a

few sequences harboring common subsequences in the size of the PQS tested.

Note, that most viruses mutate much faster than bacteria, eucaryotes, or archaea,

so chances are, that subsequences of length of about 30 develop independently.

Especially if the other hypothesis is that dsDNA and +strand ssRNA viruses are

somehow related and have conserved these small pieces of subsequence during

evolution.

5. For the conservation of the PQS found in SARS-CoV-2 reference and the remaining

3297 SARS-CoV-2 genomes, please also state the overall sequence identity. Otherwise,

simply showing that the PQS are conserved by 98.6%+-7.4% renders it difficult to

assess whether this is expected, or unexpected.

6. Suppl. Figure 17 and related text. The examples given do not seem well chosen.

First, the linker sequences between the G-runs are quite large compared to the number

of layers (2). So I'd assume that, if at all, the resulting quadruplexes would be

exceptionally weak. Second, the regions of Cluster 1 and Cluster 2 are located

in the 5'UTR which is know to be well structured and conserved throughout all

betacoronavirus and even the remaining coronaviruses. In particular, Cluster 2

overlaps the well annotated SL5C which is required for replication. Cluster 2

resides in a highly complementary region that is known to form well conserved

secondary structure. This might also explain the low conservation of 27% among the

coronaviruses, if I'm not mistaken, since secondary structure, not G-quadruplex

seems to play the most important role here. The authors should therefore relate

their findings to known annotation of SARS-CoV-2.

Along with that, the authors should add to their discussion a paragraph about

the reliability of their predictions, if such can be assessed. Especially

the PQS (and PiMS) with just 2 layers and/or bulges are known to be less

stable, thus potentially do not form at all (in vitro or in vivo). Moreover,

even when NMR and CD measurements of the PQS suggest their formation, they

still have to compete against regular secondary structure formation when they

reside in their viral sequence context. The authors should elaborate on that

problem, at least in the discussion.

General remarks:

- There are multiple occurrences of ANR instead of RNA throughout the manuscript

and Supplement. Please correct them.

- In the 4th paragraph of the introduction, there is a 'nucleic acid sequences'

right after DNA and RNA. This is redundant, since the NA in both already stands

for Nucleic Acid

- In the second paragraph of Materials and Methods, line 112. What do the

authors attempt to convey with 'presence of known-to-form quadruplex structures

sequences'? Maybe a simple 'presence of known-to-form quadruplexes' would suffice?

Reviewer #3: Thank you for the opportunity to review the manuscript: “Exploring G- and C-quadruplex structures as potential targets for the severe acute respiratory syndrome coronavirus 2". This manuscript is presenting analyses of potential G-quadruplex-forming sequences (PQS) in the genome of SARS-CoV-2 and related viruses. It was interesting to read this, but there are several points which must be improved remarkably before its publication. The main problems are 1. Conclusions of the analyses are wrongly interpreted, 2. The various methods must be used for their results evaluation and 3. The used methods must be described in manuscript with details. Although I found this manuscript interesting the major revision is necessary before its publication. I recommend a major revision of this manuscript.

Major points:

1. Conclusion are misleading. The authors find that compare to other tested viruses, there are no potential G-quadruplex forming (PQS) sequences in SARS-cov2 genome. Then they changed the algorithm for PQS search – “lax” configuration – with loop size 20bp – and claimed that these PQS could be may be suitable targets. This approach is wrong. The stability of these G-quadruplexes will be extremely low, so their formation in vivo is very doubtful. The right conclusion will be that compare to other genomes – the G-quadruplexes are probably very rare (if presented) in SARS-cov2.

2. The alternative algorithms must be used as a control and discussed (for example QGRS Mapper and G4Hunter web are freely available and easily accessible for quick evaluations).

3. The Title should be improved, the authors did not target any structures, just predicted them.

4. Description of experimental methods in Material and Method section is missing.

5. How the PQS score is set? What these number means? Please compare with G4Hunter score and QGRS score.

6. The comparison of PQS in real genomes must be compared statistically with scrambled/random sequences – it is possible than the same amount or maybe more PQS will be in scrambled sequence. With this control your analyses do not prove anything.

7. Do not use suggestion the the Result part. Just described your results (e.g. line 231: Here they may play their biological role if formed., line: 266, 277: „may be“. We are not interested what „may be“ in Results, but we would like to see the facts only. Keep your theories and “maybe” into Discussion part.

8. Figure 2B. Compare results with already in vivo proved sequences.

9. Figure 2C, Show the same result with confirmed iM and compare the spectra.

Minor points:

1. Please polish the terminology. I-motifs are formed by four strands, however to use C-quadruplex is unusual, because the structure is completely different from G-quadruplexes. Fours strand structures: G-quadruplexes and i-motifs.

2. Please do not repeat the same first sentence in Methods and Results section.

3. Line 195 and 196 – do not discuss in Result section.

4. Line 294 and 295 – again this is not your result so move to discussion part

5. Line 316 and 317 – I do not understand. How your results on RNA viruses proved “the especially for DNA, G4-iM Grinder can be used”? Any algorithm can be used if you used “lax” settings.

6. PLOS authors have the option to publish the peer review history of their article (what does this mean?). If published, this will include your full peer review and any attached files.

Reviewer #1: No

Reviewer #2: No

Reviewer #3: No

---

## [Author Response · Author response to Decision Letter 0]

24 Feb 2021

A rebuttal letter that responds to each point raised by the academic editor and reviewer has been submitted.

A marked-up copy of the manucripst that highlights changes made to the original version has been submitted.

An unmarked version of the revised paper without tracked changes has been submitted.

Figures that comply with PACE, have been submitted within the RAR file.

The supplementary material has been submitted.

---

## [Decision Letter · Decision Letter 1]

25 Mar 2021

PONE-D-20-34438R1

Potential G-quadruplexes and i-Motifs in the SARS-CoV-2

PLOS ONE

Dear Dr. Belmonte Reche,

Thank you for submitting your manuscript to PLOS ONE. After careful consideration, we feel that it has merit but does not fully meet PLOS ONE’s publication criteria as it currently stands. Therefore, we invite you to submit a revised version of the manuscript that addresses the points raised during the review process.

In particular, one of the referees notes a technical issue with the way that you have performed your analysis on random 16-nt long sequences and the probability that they would form an exact match. Specifically,

"The authors matched PQS and PiMS found in SARS-CoV-2 in other viruses and claim that these findings 'cannot be explained by chance'."

Although you have addressed this in your response, the second referee notes that, because of SARS-Cov2 genome has a length of > 25000 nt, this presents multiple chances for alignment, while your analysis only gives statistics when comparing 16 nt against 16 nt (not 16nt against 25,000+), and thus the referee is concerned that the presence of the PQS and PIMS are more common then you expect. I would be glad to re-consider your revised manuscript after you have responded to this issue.

We look forward to receiving your revised manuscript.

Kind regards,

Eric Charles Dykeman, Ph.D.

Academic Editor

PLOS ONE

Reviewers' comments:

Reviewer's Responses to Questions

**Comments to the Author**

1. If the authors have adequately addressed your comments raised in a previous round of review and you feel that this manuscript is now acceptable for publication, you may indicate that here to bypass the “Comments to the Author” section, enter your conflict of interest statement in the “Confidential to Editor” section, and submit your "Accept" recommendation.

Reviewer #2: All comments have been addressed

Reviewer #3: All comments have been addressed

2. Is the manuscript technically sound, and do the data support the conclusions?

Reviewer #2: Yes

Reviewer #3: No

3. Has the statistical analysis been performed appropriately and rigorously? 

Reviewer #2: N/A

Reviewer #3: No

4. Have the authors made all data underlying the findings in their manuscript fully available?

Reviewer #2: Yes

Reviewer #3: Yes

5. Is the manuscript presented in an intelligible fashion and written in standard English?

Reviewer #2: Yes

Reviewer #3: Yes

6. Review Comments to the Author

Reviewer #2: The authors revised their manuscript and properly addressed all the comments and questions raised in my previous report.

Reviewer #3: 1. The experiment with the "shuffled" or "random" sequence was preformed wrongly. :

"To complement the analysis, we randomly shuffled sequences (16 nucleotides long)."

Please take complete virus sequence then shuffle and analyse this 29903 nt long sequence - and compare the number of PQS sequences in the real virus sequence and "shuffled" sequence.

2. The abstract do not correspond to the results. It was found that there are less PQS in Coronaviridae compare another viruses. This fact must be part of the abstract and disscussed accordingly.

3. "hundreds of potential quadruplex candidates were discovered" - please use exact numbers and significance in the abstract statements.

7. PLOS authors have the option to publish the peer review history of their article (what does this mean?). If published, this will include your full peer review and any attached files.

Reviewer #2: No

Reviewer #3: No

---

## [Author Response · Author response to Decision Letter 1]

1 Apr 2021

Please find the response to Reviewer 3 second comments attached on submission (as it has graphs which cannot be just pasted here).

---

## [Decision Letter · Decision Letter 2]

12 Apr 2021

Potential G-quadruplexes and i-Motifs in the SARS-CoV-2

PONE-D-20-34438R2

Dear Dr. Belmonte Reche,

We’re pleased to inform you that your manuscript has been judged scientifically suitable for publication and will be formally accepted for publication once it meets all outstanding technical requirements.

Kind regards,

Eric Charles Dykeman, Ph.D.

Academic Editor

PLOS ONE

Additional Editor Comments (optional):

Reviewers' comments:

Reviewer's Responses to Questions

**Comments to the Author**

1. If the authors have adequately addressed your comments raised in a previous round of review and you feel that this manuscript is now acceptable for publication, you may indicate that here to bypass the “Comments to the Author” section, enter your conflict of interest statement in the “Confidential to Editor” section, and submit your "Accept" recommendation.

Reviewer #3: All comments have been addressed

2. Is the manuscript technically sound, and do the data support the conclusions?

Reviewer #3: Yes

3. Has the statistical analysis been performed appropriately and rigorously? 

Reviewer #3: Yes

4. Have the authors made all data underlying the findings in their manuscript fully available?

Reviewer #3: Yes

5. Is the manuscript presented in an intelligible fashion and written in standard English?

Reviewer #3: Yes

6. Review Comments to the Author

Reviewer #3: Thank you. Authors have improved the manuscript and added requested analyses. I recommend this manuscript for publication.

7. PLOS authors have the option to publish the peer review history of their article (what does this mean?). If published, this will include your full peer review and any attached files.

Reviewer #3: No

---

## [Editor Report · Acceptance letter]

27 May 2021

PONE-D-20-34438R2 

Potential G-quadruplexes and i-Motifs in the SARS-CoV-2 

Dear Dr. Belmonte-Reche:

I'm pleased to inform you that your manuscript has been deemed suitable for publication in PLOS ONE. Congratulations! Your manuscript is now with our production department. 

Kind regards, 

on behalf of

Dr. Eric Charles Dykeman 

Academic Editor

PLOS ONE